# An Adaptive ORB-SLAM3 System for Outdoor Dynamic Environments

**DOI:** 10.3390/s23031359

**Published:** 2023-01-25

**Authors:** Qiuyu Zang, Kehua Zhang, Ling Wang, Lintong Wu

**Affiliations:** 1College of Mathematics and Computer Science, Zhejiang Normal University, Yingbin Avenue, Jinhua 321005, China; 2Key Laboratory of Urban Rail Transit Intelligent Operation and Maintenance Technology & Equipment of Zhejiang Province, Zhejiang Normal University, Yingbin Avenue, Jinhua 321005, China

**Keywords:** visual SLAM, dynamic environment, positional estimation

## Abstract

Recent developments in robotics have heightened the need for visual SLAM. Dynamic objects are a major problem in visual SLAM which reduces the accuracy of localization due to the wrong epipolar geometry. This study set out to find a new method to address the low accuracy of visual SLAM in outdoor dynamic environments. We propose an adaptive feature point selection system for outdoor dynamic environments. Initially, we utilize YOLOv5s with the attention mechanism to obtain a priori dynamic objects in the scene. Then, feature points are selected using an adaptive feature point selector based on the number of a priori dynamic objects and the percentage of a priori dynamic objects occupied in the frame. Finally, dynamic regions are determined using a geometric method based on Lucas-Kanade optical flow and the RANSAC algorithm. We evaluate the accuracy of our system using the KITTI dataset, comparing it to various dynamic feature point selection strategies and DynaSLAM. Experiments show that our proposed system demonstrates a reduction in both absolute trajectory error and relative trajectory error, with a maximum reduction of 39% and 30%, respectively, compared to other systems.

## 1. Introduction

Simultaneous Localization and Mapping (SLAM) is a method for robots and other autonomous vehicles to map out unknown environments and simultaneously estimate their positions within those environments. To better perform tasks such as navigation, mobile robots rely on the accuracy of the map, robots mainly utilize the camera to estimate the position and create a map of the environment in Visual SLAM systems. Visual SLAM is a popular choice because of its ability to capture rich visual information. However, traditional visual SLAM systems assume a static environment and can struggle to maintain accurate localization and mapping because of dynamic objects. The research questions in this study focused on reducing trajectory errors of the visual SLAM system in a dynamic environment.

Studies on visual SLAM represent a growing field. One of the most cited studies is that Qin et al. [1] proposed the VINS-Mono in 2018 using optical flow as a front-end with high robustness at fast motion, which is widely used in unmanned aerial vehicles. However, VINS-Mono is not suitable for weak textured environments. In 2021, Koestler et al. [2] proposed TANDEM, the first time using the global TSDF model for monocular dense tracking. Perhaps the best-known study is the ORB-SLAM, which Mur-Artal et al. [3] proposed based on ORB features in 2015, Their team iterated ORB-SLAM2 [4] in 2017, optimized the algorithm performance, added support for stereo and RGB-D cameras. Campos, Carlos et al. [5] released ORB-SLAM3 in 2021. ORB-SLAM3 was the first feature-based tightly coupled visual-inertial odometry system with the atlas multi-map system, adding support for fish-eye cameras and IMUs. Most existing visual SLAM systems are subject to potential methodological weaknesses; they assume a static environment for localization and mapping. However, there are always many dynamic objects in the real scene, which in turn affects the accuracy and the robustness of the visual SLAM system [6,7,8].

Recently, considerable literature has grown up around the theme of visual SLAM systems in dynamic environments. A summary of related work with advantages and disadvantages is presented in Table 1. In the study on semantic information-based for dynamic environments SLAM, Erliang Yao et al. [9] used the transformation matrix of IMU and reprojection error to distinguish dynamic feature points.VDO-SLAM, proposed by Zhang et al. [10] in 2020, uses semantic information to estimate accurate motion, but the system requires pre-processing datasets including instance-level semantic segmentation and optical flow estimation, so the system is computationally intensive. Zhang et al. [11] used RGB-D cameras and a K-means clustering algorithm to distinguish foreground and background features in the bounding box. The DS-SLAM proposed by Chao Yu et al. [12] combined the semantic segmentation network with ORB-SLAM2 effectively reducing the trajectory error. However, the octree map must be reconstructed when loop closure and DS-SLAM are hard to run in real-time. A limitation of the above studies is that most of them only use the TUM Dynamic Objects dataset or the other indoor dataset, not demonstrating robustness in outdoor dynamic environments with complex lighting and rich features. There are a large number of published studies about ORB-SLAM-based dynamic environment visual SLAM study. DynaSlam, proposed by Bescos et al. [13] in 2018, combined MASK-RCNN semantic segmentation network with ORB-SLAM2, Maps generated from static environments will be used to fix frames that are obscured by dynamic objects, effectively reduce trajectory error in dynamic environments, however, the real-time performance of Dynaslam is poor. Bescos et al. [14] iterated DynaSlam II in 2021, which uses a single bundle adjustment that simultaneously estimates the camera pose and trajectory of moving objects. However, Dynaslam II performs poorly in low-texture environments. Wu et al. [15] proposed YOLO-SLAM based on ORB-SLAM2 using YOLOv3 to detect dynamic objects, Although YOLO-SLAM has significant performance in the TUM dataset resulting in a 98% reduction in absolute trajectory error, the algorithm does not evaluate the accuracy in dynamic outdoor environments. Liu et al. [16] combined YOLOv3 with ORB-SLAM2. The system uses GMM to build a background image model and detect the dynamic regions in the foreground. However, the accuracy of dynamic region detection can be improved. Zhang et al. [17] employed the YOLOv5 to detect dynamic information, and integrated the optical flow technique for a second detection pass to enhance detection precision. They subsequently constructed a global map by utilizing keyframes and removing highly dynamic objects. The system under examination demonstrated superior performance in terms of accuracy, as evidenced by a substantial reduction in absolute trajectory error (up to 97.8%) and relative trajectory error (59.7%), as well as a marked decrease in average tracking time (up to 94.7%). However, the study is accompanied by certain limitations, one of which is that when the dynamic objects in the environment remain stationary for an extended period, they will obscure the static background, thus resulting in an imprecise construction of the map. Zhong et al. [18] improved the accuracy of localization by weighting the feature points. However, the time consumption of the system has increased significantly in static scenes, such as TUM sitting_xyz dataset. Soares et al. [19] and Han et al. [20] retained static priori dynamic objects although they combined optical flow to determine dynamic objects, which will reduce localization accuracy. Typical methods for point feature matching can be summarized as descriptor-based and pixel-based methods [21], due to the difficulty of establishing sufficient dynamic data associations, the descriptor-based method can be challenging to apply to highly dynamic objects [22]. As such, many systems resort to using the optical flow method to address this issue [23,24,25], outliers are then removed from the essential matrix using the RANSAC method [26].

This study aims to address the issue of low accuracy in dynamic outdoor environments by proposing an adaptive feature point selection system. To this end, YOLOv5s is utilized for a priori dynamic object detection, and coordinate attention is employed to improve detection accuracy, the bounding box is deemed a dynamic region if the ratio of the infinite norm of the resulting essential matrix falls below a certain threshold. Finally, feature points in the dynamic region are removed, and stereo feature point matching use only static feature points.

To sum up, the main contributions of this work are summarized as follows:We proposed an adaptive feature point system to enhance the accuracy of ORB-SLAM systems in dynamic environments. This system utilizes YOLOv5s with coordinate attention to detect dynamic objects a priori in the frame and is combined with the LK optical flow method for accurate determination of dynamic areas. Additionally, an adaptive feature point selector is proposed to remove dynamic feature points based on the frame share and the number of a priori dynamic objects bounding boxes in the current frame. By utilizing only static feature points for mapping, the mapping error of the system in dynamic environments is reduced.In the adaptive feature point selector, we proposed a geometric method for identifying dynamic regions. We calculate the infinitesimal parametrization of the essential matrix of feature points within the bounding box and the bounding box expansion layers, and determine the dynamic region based on the ratio of the two infinitesimal parametrizations.

In Section 2, we describe the specific methods used by the system. The results obtained from the system are described in Section 3. Finally, we conclude with a short discussion in Section 4 and a conclusion in Section 5.

## 2. Materials and Methods

In this section, we will describe the procedures and methods used in this system. First, the general flow of the system is described. Then, the YOLOv5s with coordinate attention will be presented. Finally, the adaptive feature point selector will be introduced.

### 2.1. System Framework

The proposed system is based on traditional ORB-SLAM3. ORB-SLAM3 contains three threads: Tracking, Local Mapping, and Local Closing. We added the YOLOv5s Detection thread and LK optical flow thread on top of ORB-SLAM3. The system framework for the proposed system is depicted in Figure 1.

Frames will be constructed after system initialization is completed in the Tracking thread in ORB-SALM3. Among the existing improved system based on ORB-SLAM, dynamic point deletion is generally performed in the Tracking thread [27], dynamic feature points deleted after feature point extraction [28,29,30]. In our system, the YOLOv5s thread subscribes to left-frame images and performs detection of a priori dynamic objects, sending ROS messages with bounding box coordinates. The adaptive selector subscribes to these ROS messages and checks whether any feature points are located within the bounding box. Based on the number of a priori dynamic objects and the percentage of the frame occupied by a priori dynamic objects in the current frame, the adaptive selector then determines whether to remove these feature points. We set *N* is the number of a priori dynamic objects in the current frame and *P* is the percentage of the frame occupied by a priori dynamic objects in the current frame. The threshold to remove all feature points in the bounding boxes is set to *T*. If N<T and P<50%, all feature points within the bounding box are deleted. On the other hand, if N>T or P>50%, the positions of the feature points within the bounding box between the current frame and the previous frame are tracked by Lucas-Kanade optical flow. The essential matrix of the feature points in each bounding box and each bounding box expansion between the current frame and the previous frame will be calculated. We set the essence matrix of one of the bounding boxes between the two frames is $E_{u}$ and the essential matrix of its corresponding bounding box expansion layer is $E_{v}$. The RANSAC method is used to reduce the error of essence matrices $E_{v}$ and $E_{u}$. The infinite norm $L_{1}$ of $E_{u}$ and the infinite norm $L_{2}$ of $E_{v}$ will be separately calculated, and the bounding box is considered a static region by comparison $L_{2}$ and $L_{1}$. Stereo matching is only performed using feature points from static regions. Once static frames have been generated, active map filters new keyframes, and the map is optimized.

### 2.2. YOLOv5 with Coordinate Attention

YOLOv5 thread subscribes to the original left-frame image and detects a priori dynamic objects. The YOLOv5 Detection thread is added to the system to detect a priori dynamic objects in the current frame. Adjusting the dynamic feature weights of the input image is called the attention mechanism [31]. Coordinate attention is a network that incorporates location information into channel attention [32].

For any intermediate feature tensor in the network $X=\left[x_{1},x_{2},...,x_{c}\right]\in R^{C*H*W}$, the spatial information is first embedded by averaging global pooling kernel along the width and height directions using (H,1) and (1,W), respectively. The output of the *c*-th channel for height *h* can be expressed as $z_{c}^{h}$, As shown in Equation (Equation 1). The output of the *c*-th channel for width *w* can be expressed as $z_{c}^{w}$, as shown in Equation (Equation 2).
(1)$$z_{c}^{h}\left(h\right)=\frac{1}{W}\underset{0\leq j<W}{\sum^{}}\;x_{c}\left(h,j\right)$$
where *W* represents the width of the feature tensor, $z_{c}^{h}$ represents the squeeze step for the *c*-th channel with height direction encode information, *h* is the height of the feature map, *c* represents the number of channels in the feature map.
(2)$$z_{c}^{w}\left(h\right)=\frac{1}{H}\sum_{0\leq i<H}^{}x_{c}\left(i,w\right)$$
where *H* represents the height of the feature tensor, $z_{c}^{w}$ represents the squeeze step for the *c*-th channel with width direction encode information, *w* is the width of the feature map.

After that, as shown in Equation (Equation 3), The feature maps representing width and height are concatenated and subsequently fed into the 1∗1 convolution module, and the batch normalization feature map F1 is fed into the activation function *S* to obtain feature map f.
(3)$$f=S\left(F_{1}\left(\left[z^{h},z^{w}\right]\right)\right)$$
where f represents the intermediate feature map with spatial information, S is a non-linear activation function.

Subsequently, the attention weights in the height direction, represented as $g^{h}$, are computed through the application of a sigmoid activation function σ, as detailed in Equation (Equation 4). Attention weights in the width direction, denoted as $g^{w}$, are obtained through an analogous procedure. The final output of the coordinate attention $y_{c}$ can be formulated according to Equation (Equation 5). The network model with the addition of coordinate attention is shown in Figure 2.
(4)$$g^{h}=\sigma \left(F_{h}\left(f^{h}\right)\right)$$
where $g^{h}$ represents attention weights for height direction. σ is the Sigmoid function. $F_{h}$ and $f^{h}$ represents two separate tensors about *f* along the spatial dimension.
(5)$$y_{c}\left(j.i\right)=x_{c}\left(j,i\right){*g}_{c}^{h}\left(j\right){*g}_{c}^{w}\left(i\right)$$
where $g^{w}$ represents attention weights for width direction. $y_{c}$ denotes the attention weight with spatial information for the *c*-th channel.

The loss function of YOLOv5 consists of three components: classification loss, confidence loss, and localization loss. The classification loss and the confidence loss is calculated using cross-entropy loss. The localization loss is calculated using CIoU loss, which improves the accuracy of object detection by taking into account more information about the geometric relationship and the shape of the bounding boxes.

### 2.3. Adaptive Feature Points Selector

In a dynamic environment, eliminating dynamic feature points can significantly increase the accuracy of the visual SLAM systems. However, simply removing all feature points associated with dynamic objects in complex outdoor settings can lead to a reduction in accuracy due to the limited number of remaining feature points. To address the issue of reduced accuracy in complex outdoor settings due to dynamic objects, we propose an adaptive feature point selection method that selectively filters out dynamic feature points while attempting to preserve as many static feature points as possible.

After receiving ROS messages from YOLOv5 containing the coordinates of a priori dynamic objects, the adaptive feature point selector (shown in Algorithm 1) determines whether to retain or eliminate feature points based on the number of a priori dynamic objects and percentage of a priori dynamic objects occupied in the current frame. It checks if each feature point is within the bounding box in the current frame. If the feature point is not located in the bounding box, it is considered a static feature point and retained for stereo matching. If the feature point is located in the bounding box, the adaptive selector is used to determine whether it should be retained or eliminated. If the percentage of a priori dynamic objects occupied in the frame is greater than 50%, the system uses a geometric method to determine whether the a priori dynamic object is static or not. If the percentage of a priori dynamic objects occupied in the frame is less than 50%, the system checks if the number of a priori dynamic objects is greater than a predetermined threshold value. In our study, we varied the a priori dynamic object threshold and observed that utilizing a threshold of 5 led to a decrease in the absolute trajectory error within the proposed system. Therefore, the threshold value is set to 5. If the number of a priori dynamic objects is less than the threshold, the feature points within the bounding box are not used for mapping. The geometric method is used to determine whether the dynamic object is static or not while the number of a priori dynamic objects is greater than the threshold. When using the geometric method, the system adds an expansion layer of 20 pixels outside each bounding box and the position of feature points within the bounding box and expansion layer between the current frame and the previous frame are tracked by LK optical flow. If there are no feature points in the expansion layer, the dynamic object is considered to be in a low feature point environment, and the feature points within the bounding box are used to build the map. First, for all feature points in a bounding box in the current frame and their corresponding feature points in the previous frame form a set $P_{1}$. Randomly select the four matching pairs of feature points in $P_{1}$ form a set $P_{2}$. For a pair of matched feature points in set $P_{2}$, there is a feature point $X_{1}$ located in a bounding box in the current frame, and there is a feature point $X_{2}$ matching it in the previous frame. The essential matrixes of the bounding box are solved by the Eight-point-algorithm. Suppose the normalized coordinates of $X_{1}$ is $x_{1}={\left[u_{1},v_{1},1\right]}^{T}$ and the normalized coordinates of $X_{2}$ is $x_{2}={\left[u_{2},v_{2},1\right]}^{T}$. For an essential matrix $E_{1}=\left[\begin{array}{ccc} e_{11} & e_{21} & e_{31}\\ e_{12} & e_{22} & e_{32}\\ e_{13} & e_{23} & e_{33} \end{array}\right]$, According to the epipolar constraint we can obtaine Equation (Equation 6), As shown in the Equation (Equation 7), Expressing epipolar constraint as a linear form. Four randomly selected pairs of matching points to form a system of eight linear equations, which gives the linear form to obtain the essential matrix $E_{2}$.
(6)$$\left(u_{2},v_{2},1\right)\left[\begin{array}{ccc} e_{11} & e_{21} & e_{31}\\ e_{12} & e_{22} & e_{32}\\ e_{13} & e_{23} & e_{33} \end{array}\right]\left(\begin{array}{c} u_{1}\\ v_{1}\\ 1 \end{array}\right)=0$$
where $u_{1}$, $v_{1}$ are the normalized spatial coordinate of the point $x_{1}$, $u_{2}$, $v_{2}$ are the normalized spatial coordinate of the point $x_{2}$, $\left[\begin{array}{ccc} e_{11} & e_{21} & e_{31}\\ e_{12} & e_{22} & e_{32}\\ e_{13} & e_{23} & e_{33} \end{array}\right]$ represents essential Matrix $E_{1}$.
(7)$$\left[u_{2}u_{1,}u_{2}v_{1,}u_{2,}v_{2}u_{1,}v_{2}v_{1,}v_{2},u_{1,}v_{1,}1\right]\left(\begin{array}{c} e_{11}\\ e_{21}\\ e_{31}\\ e_{12}\\ e_{22}\\ e_{23}\\ e_{13}\\ e_{23}\\ e_{11} \end{array}\right)=0$$
where $e={\left[e_{11},e_{21},e_{31},e_{12},e_{22},e_{32},e_{13},e_{23},e_{33}\right]}^{T}$ denotes the vector form of $E_{1}$.
**Algorithm 1:** Adaptive dynamic feature point selection strategy.**Input:** Original feature point, Bounding box**Output:** Static feature point 1: **if** feature points in the boundingbox **then** 2:    Use LK optical flow to match a prior dynamic objects between frames 3:    **if** a priori dynamic object screen share ≥ 50% **then** 4:      Determine if the number of a priori dynamic objects is greater than the threshold 5:      **if** number of a prior dynamic objects ≥ 5 **then** 6:          Calculate essential matrix of prior dynamic region bwtween frame 7:          Calculating infinite norms of essential matrix 8:          Calculate the essential matrix of the expansion layer bwtween frame 9:          Calculating infinite norms of essential matrix10:        **if** ratio of two infinite norm is greater than the threshold value **then**11:            Delete dynamic feature points12:        **else**13:          ORB matches14:        **end if**15:      **end if**16:    **end if**17: **end if**18: **return** Outputs

Then, select feature points in $P_{1}$ that are not in $P_{2}$. The distance from these feature points to each epipolar lines in pixels will be calculated. Our experimental results have revealed that the system exhibits a reduced absolute trajectory error when the pixel distance from the matching point to the epipolar line is 1 pixel. Therefore in this paper, we set the threshold to one pixel. If the maximum distance is greater than one pixel, the pair is considered an outlier. If the maximum distance is less than a threshold value, the pair is considered an inlier. The number of inliers will be recorded. Finally, The algorithm will reselect four pairs of matching points from the set $P_{1}$ and calculate the number of inliers. When the threshold *t* is reached after a number of iterations, the essential matrix with the most inliers is chosen. The essential matrix for the bounding box expansion layer is also obtained through this process.

After obtaining the essence matrix $E_{u}$ in the bounding box and the essence matrix $E_{v}$ in the expansion layer. Use $∥x_{\infty }∥=\max_{1\leq i<8}\left|x_{i}\right|$ find the infinite parametric $L_{1}$ of $E_{u}$ and the infinite parametric $L_{2}$ of $E_{v}$. Calculate the ratio *r* using $r=\frac{\min(L_{1},L_{2})}{\max(L_{1},L_{2})}$, If *r* is greater than the threshold 0.95, the a priori dynamic object in the bounding box is considered as static. If *r* is less than the threshold 0.95, the adaptive selector sets the pixel coordinates of the left-frame feature point in the bounding box to (−1,−1) and performs stereo feature points matching after completing filtering of all feature points in the current frame.

With the above approach, our system has higher accuracy in dynamic environments compared to ORB-SLAM3. The results obtained from the system are described in the next chapter.

## 3. Results

To further evaluate our system, We employed the KITTI odometry dataset [33] as an evaluation benchmark. The KITTI dataset is a widely-used resource for evaluating the accuracy of visual SLAM systems in dynamic environments. KITTI odometry is an open-source dataset containing urban, highway, and other scenes, and the dataset contains ground truth which allows us to evaluate the trajectory accuracy.

To further assess the effectiveness of the adaptive dynamic feature point selection strategy proposed in this paper, we compare it to two other algorithms: one that removes all feature points located within bounding boxes and one that uses only the geometric method described above, but without the adaptive feature point selector. The accuracy of the system is evaluated using the EVO and KITTI development kits, and the evaluation metrics used are the absolute trajectory error (APE) and the relative positional error (RPE) [34]. The relative positional error represents the error between the algorithm-generated trajectory and the true positional trajectory at fixed intervals, while the absolute trajectory error reflects the overall difference between the algorithm-generated trajectory and the true trajectory.

We combine the classes in the KITTI object into three: Car, Pedestrian, and Cyclist. The KITTI dataset is divided into a training set, a validation set, and a test set according to 8:1:1, and ablation experiments are conducted on the validation set. We performed 300 epochs, batch-size set to 128, and also used adaptive anchor frames and adaptive image scaling. The recognition results of YOLOv5s with different attention mechanisms in the KITTI dataset are shown in Table 2. MAP@0.5 improved by 1% after adding coordinate attention. The test set and validation set were trained together and validated using the test set, and the results are shown in Table 3.

The proposed system was utilized to conduct threshold experiments for matching point to epipolar line distances on the KITTI 04 dataset, which comprises a greater number of a priori dynamic objects, and the KITTI 06 dataset, which comprises a higher quantity of static a priori dynamic objects. The a priori dynamic object threshold was established at 5. As shown in Table 4, Experiments show that the system’s absolute trajectory error is reduced when the pixel distance threshold from the matching point to the epipolar line is set at 1 pixel. In the adaptive feature point selector, the threshold selection for a priori dynamic objects was evaluated through experimentation using various threshold values. As shown in Table 5, the results revealed that the system displayed decreased trajectory errors when the threshold was set to a value of 5.

To evaluate the performance of ORB-SLAM3, the algorithm for deleting all feature points within bounding boxes, the geometric method, and our proposed method, we use the KITTI2bag tool to convert the dataset to ROSbag format and test each system on it. First, we convert the KITTI trajectory file with ground truth into TUM trajectory files, as the KITTI trajectory generated by ORB-SLAM3 does not contain timestamps. Then, we use SE(3) Umeyama alignment while processing panning rotation and scale. Each system is run 5 times on each dataset. Since the results of the system are random, we remove the maximum and minimum values from the five runs, average the remaining three results, and compare the trajectory errors of the median results. The hardware platform used for this experiment is a laptop with an NVIDIA GeForce RTX 3060 Laptop, AMD Ryzen 5900hx, 16G RAM, and Ubuntu 20.04 OS. The absolute trajectory error was evaluated using EVO, and the results are shown in Table 6. The relative positional error was evaluated using the KITTI development kit, and the results are shown in Table 7. The best results are bolded. Unfortunately. We did not get the KITTI 03 and 08 dataset, so only the remaining dataset was used for the experiment.

In contrast to the ORB_SLAM3 system, our proposed system has been shown to exhibit a substantial reduction in the absolute and relative trajectory error. Empirical evaluations reveal that the absolute trajectory error is decreased by a maximum of 39%, and the relative trajectory error is diminished by a maximum of 30%. Compared to the geometric method, our proposed method significantly reduces the absolute trajectory error on datasets with slow-moving objects, such as the 05 and 06 datasets. 05 dataset is a vehicle driving in an urban environment, which contains a large number of stationary vehicles, moving vehicles, walking people, etc. As shown in Figure 3, our system has a smaller median error on the 05 dataset compared to the geometric method. The 06 dataset is a vehicle driving in a circle in the city. In the context of this dataset, our proposed system has demonstrated a reduction in the relative trajectory error per 100 meters of 30.6%, as well as a reduction in the absolute positional error of 17%, and the trajectory of our system is closer to the true value of the trajectory on the x-axis, as shown in Figure 4.

Compared with the algorithm of deleting all dynamic points in the bounding box, our system can significantly reduce the relative positional error in urban environments with a large number of static a priori dynamic objects, such as the 01, 02, 06, 09 datasets. In our evaluation, the relative trajectory error per 100 meters is seen to be decreased by 30% on the 02 data set. For example, there is a significant reduction in relative positional error on the KITTI 01 dataset, as shown in Figure 5.

We compared the state-of-the-art DynaSLAM (Geometric methods) using SE(3) Umeyama alignment, and the results are shown in Table 8, with the best results bolded. Compared to DynaSLAM, our system has reduced absolute trajectory errors on the five datasets. As shown in Table 9, our system not only enhances precision but also preserves an acceptable level of real-time functionality, In comparison to ORB_SLAM3, our system exhibits an improvement in mean total tracking time as a result of the incorporation of dynamic feature point detection. However, the advancement in mean total local mapping time is not substantial, as the dynamic feature points are eliminated directly within the tracking thread. In contrast to the current state-of-the-art DynaSLAM, our system demonstrates exceptional real-time performance. The enhanced performance of ORB_SLAM3 allows for a substantial increase in efficiency for tracking threads in our system, as demonstrated by a 50% improvement per frame when compared to the DynaSLAM system.

## 4. Discussion

To improve the accuracy and robustness of visual SLAM in dynamic environments, we propose an adaptive ORB-SLAM3 system for outdoor dynamic environments. The system employs an adaptive feature point selection strategy to select suitable feature points for map building, a RANSAC-based geometric method to identify dynamic regions, and only uses static feature points for map building, thereby reducing the trajectory error. Experiments on the KITTI dataset show that our system can significantly reduce the trajectory error of the visual SLAM system in a dynamic environment compared to traditional methods and other feature point selection strategies. An evaluation of our system in comparison to other systems reveals a significant reduction in both the absolute and relative trajectory error. Specifically, compared to ORB_SLAM3, the absolute trajectory error is decreased by a maximum of 39%, and the relative trajectory error is diminished by 30%. The proposed system exhibits exceptional real-time performance, as demonstrated by its ability to process frames in the tracking thread with a significant reduction in the time per frame when compared to the benchmark system, DynaSLAM, specifically, consuming half the time per frame.

The implementation of an LK optical flow thread in the system has the effect of increasing its computational demands. To address this issue and optimize the system’s performance, we plan to utilize LK optical flow for feature point detection in common frames in order to decrease computational effort. Additionally, we will leverage ORB-SLAM to detect feature points in keyframes, thereby enhancing localization accuracy. In future research, we will also investigate the problem of uneven distribution of feature points following the removal of dynamic feature points. We are conducting research on dynamic environment dense map building based on this research, so the code is not open source.

## 5. Conclusions

In this paper, we present an adaptive ORB-SLAM3 system designed for outdoor dynamic environments. It utilizes YOLOv5 with coordinate attention to identify a priori dynamic objects, such as cars, people, and bicycles. The ORB-SLAM thread subscribes to ROS messages sent by YOLOv5 and removes the feature points of dynamic objects by considering the number and percentage of a priori dynamic objects occupied in the current frame. To track feature points between frames, we use LK optical flow and introduce a bounding box expansion layer to capture background motion.

We conducted experiments on the KITTI dataset and evaluated ORB-SLAM3 with two different feature point selection strategies: the algorithm that removes all dynamic points within bounding boxes and the geometric method. We also compared our system to the current state-of-the-art DynaSLAM method, and a system time consumption comparison was performed. The results of these experiments demonstrate the effectiveness of our proposed approach, in comparison to other systems, our system demonstrates a substantial improvement in terms of both absolute and relative trajectory error. Specifically, the absolute trajectory error is reduced by as much as 39% and the relative trajectory error by up to 30%. The proposed system demonstrates outstanding real-time performance capabilities.

## Figures and Tables

**Figure 1 sensors-23-01359-f001:**
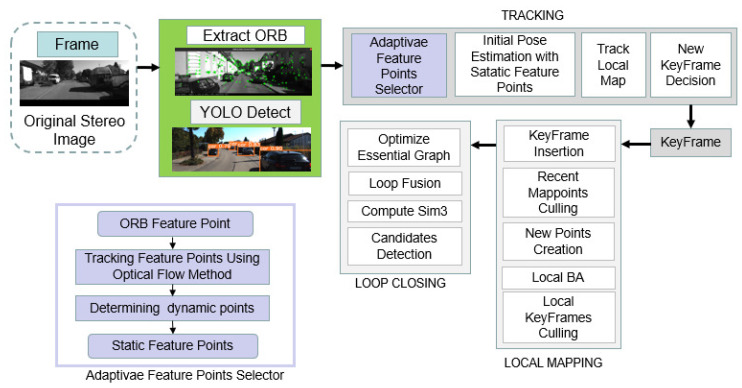
The system framework for the proposed algorithm.

**Figure 2 sensors-23-01359-f002:**
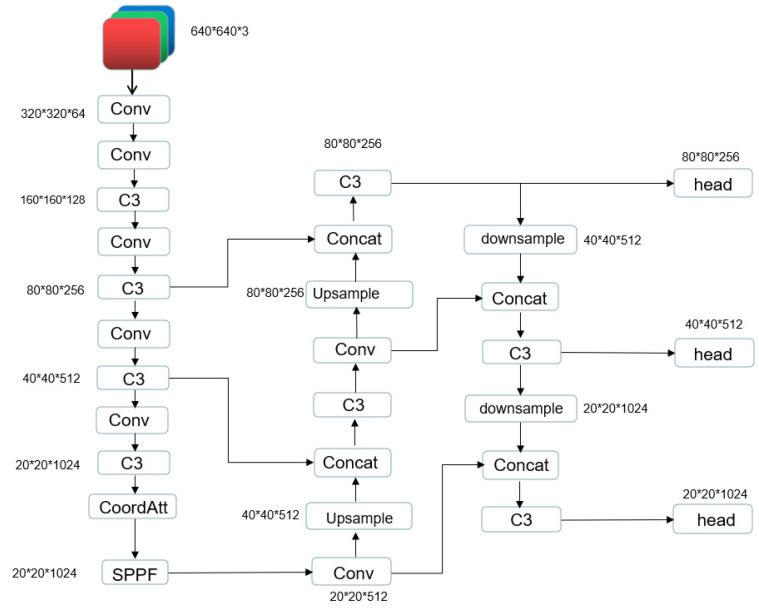
YOLOv5s network model with Coordinated Attention.

**Figure 3 sensors-23-01359-f003:**
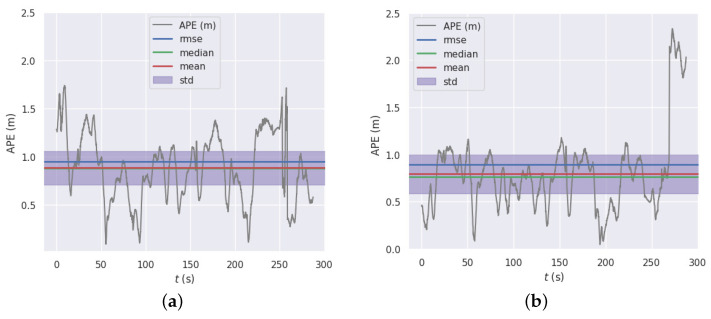
KITTI 05 dataset APE with regard to translation part(m) (with Sim(3) Umeyama alignment, while processing panning rotation and scale): (**a**) geometric method; (**b**) proposed algorithm.

**Figure 4 sensors-23-01359-f004:**
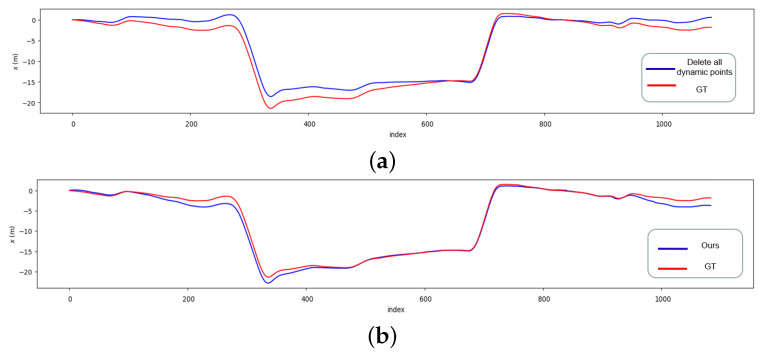
KITTI 06 dataset X-axis trajectory error, (**a**) Delete all dynamic points, (**b**) Ours.

**Figure 5 sensors-23-01359-f005:**
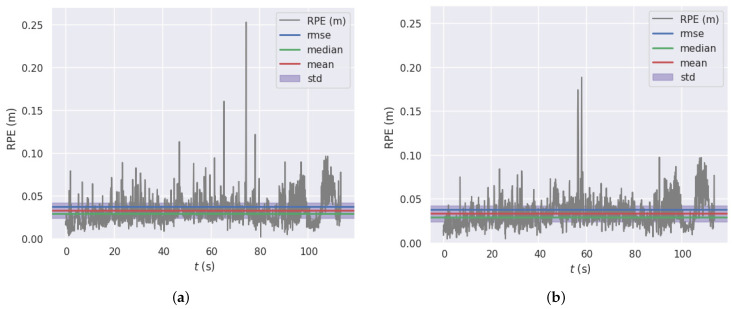
KITTI 01 dataset RPE with regard to translation part(m) for delta = 1 (frames) using consecutive pairs (with Sim(3) Umeyama alignment, while processing panning rotation and scale): (**a**) delete all dynamic points; (**b**) ours.

**Table 1 sensors-23-01359-t001:** Summary of related work with advantages and disadvantages.

System	Advantages	Disadvantages
VDO-SLAM [10]	VDO-SLAM uses semantic information for accurate motion estimation and tracking of dynamic rigid objects, without prior knowledge of shape or models.	A significant quantity of dynamic objects in the scene increases system computational demands.
PLD-SLAM [11]	PLD-SLAM improves trajectory estimation in dynamic environments by using point and line features and two consistency check techniques to filter dynamic features.	Identifying potential dynamic objects incurs significant computation time.
DS-SLAM [12]	DS-SLAM reduces trajectory errors by combining semantic segmentation networks and motion consistency checking and generates a dense semantic octree map.	Upon detection of loop closure, the octree map must be reconstructed.
DynaSLAM [13]	Dynaslam constructs an static scene map by repairing parts of the background that have been obscured by dynamic objects. Trajectories of the system exhibit a very high degree of precision.	The performance of the system is hindered by the utilization of a CNN network, resulting in a lack of real-time capability.
Zhang et al.’s system [17]	Use YOLOv5 with optical flow method to determine dynamic objects and use key frames to build maps, reducing trajectory errors.	Dynamic objects that are stationary for a long time can cause background blurring and thus cause.

**Table 2 sensors-23-01359-t002:** The recognition results of YOLOv5s with different attention mechanisms in KITTI dataset.

Model	AP			mAP@0.5
	Car	Pedestrian	Cyclist	
YOLOv5	0.963	0.82	0.835	0.873
YOLOv5 with				
Coordinate Attention	0.961	0.826	0.862	0.883
YOLOv5 with				
transformer	0.962	0.824	0.853	0.879
YOLOv5 with				
transformer and CA	0.958	0.805	0.863	0.875
YOLOv5 with				
C3SE	0.960	0.803	0.862	0.875
YOLOv5 with				
CBAM	0.963	0.818	0.852	0.877
YOLOv5 with				
CBAM and C3SE	0.960	0.804	0.875	0.879

**Table 3 sensors-23-01359-t003:** The recognition results of YOLOv5s with Coordinate Attention in KITTI dataset.

Model	AP			mAP@0.5
	Car	Pedestrian	Cyclist	
YOLOv5 with				
Coordinate Attention	0.959	0.824	0.849	0.878

**Table 4 sensors-23-01359-t004:** Comparison of different pixel thresholds for calculating the distances to the epipolar lines when calculating the essential matrix.

Sequences	1 Pixel	2 Pixel	3 Pixel
04	0.19	0.20	0.21
06	0.73	1.16	1.19

**Table 5 sensors-23-01359-t005:** Absolute trajectory error in our system using different present dynamic object thresholds in the adaptive feature point selector, The best results are bolded.

Sequences	Threshold 3	Threshold 4	Threshold 5	Threshold 6
00	2.443	2.717	**1.630**	1.650
01	**3.522**	3.979	5.291	3.988
02	**4.605**	7.057	5.572	7.453
04	0.204	0.207	**0.198**	0.277
05	1.160	1.425	**1.107**	1.498
06	1.209	1.270	**1.185**	1.197
07	0.640	**0.627**	0.628	0.709
09	1.249	**1.209**	1.241	1.236
10	1.372	1.441	**1.278**	1.593

**Table 6 sensors-23-01359-t006:** Absolute trajectory error of the original algorithm, the algorithm of removing all feature points falling in the bounding box, the geometric method using RANSAC, and the improvement algorithm(with Sim(3) Umeyama alignment), while processing panning rotation and scale.

Sequences	ORB_SLAM3	Delete All Dynamic Points	Geometric Method	Ours
00	1.92	1.69	2.28	**1.64**
01	6.31	5.94	**4.86**	6.72
02	5.41	6.37	**5.04**	5.35
04	0.22	0.22	0.2	**0.19**
05	0.99	1.02	1.01	**0.94**
06	1.2	0.88	0.92	**0.73**
07	0.63	**0.5**	0.63	0.62
09	**0.99**	1.23	1.18	1.17
10	1.46	**1.2**	1.47	1.25

**Table 7 sensors-23-01359-t007:** The relative positional error of the original algorithm, the algorithm of removing all feature points falling in the bounding box, the geometric method using RANSAC, and the improvement algorithm(with Sim(3) Umeyama alignment), while processing panning rotation and scale.

	ORB_SLAM3		Delete All Dynamic Points		Geometric Method		Ours	
Sequences	RPE[%]	RPE[°/100M]	RPE[%]	RPE[°/100M]	RPE[%]	RPE[°/100M]	RPE[%]	RPE[°/100M]
00	5.23	1.85	5.82	2.02	**5.04**	1.78	5.07	**1.76**
01	1.6	0.36	1.6	0.35	1.57	**0.33**	**1.55**	0.35
02	**1.82**	**0.46**	2.47	0.83	2.4	0.68	2.01	0.58
04	1.65	0.2	1.65	0.21	1.65	0.18	**1.54**	**0.14**
05	1.21	0.41	**1.09**	**0.31**	1.65	0.57	1.37	0.44
06	1.41	0.32	1.45	**0.25**	1.41	0.36	**1.27**	**0.25**
07	1.48	0.58	**1.1**	**0.36**	1.24	0.47	1.5	0.58
09	**1.42**	**0.39**	1.63	0.47	1.6	0.45	1.48	**0.39**
10	1.48	0.56	1.46	**0.54**	1.47	0.58	**1.44**	0.55

**Table 8 sensors-23-01359-t008:** Comparison of absolute trajectory error with DynaSLAM.

Sequences	ORB_SLAM3	DynaSLAM	Ours
00	1.92	**1.3**	1.64
01	**6.31**	10.47	6.72
02	5.41	5.73	**5.35**
04	0.22	0.22	**0.19**
05	0.99	**0.83**	0.94
06	1.2	0.76	**0.73**
07	0.63	**0.52**	0.62
09	**0.99**	3.08	1.17
10	1.46	**1.05**	1.25

**Table 9 sensors-23-01359-t009:** Real-time comparison of ORB_SLAM3, our system, and Dyna SLAM.

	ORB_SLAM3		Ours		Dyna Slam
**Sequences**	**Mean Total Tracking Time [ms]**	**Mean Total Local MAPPING Time [ms]**	**Mean Total Tracking Time [ms]**	**Mean Total Local Mapping Time [ms]**	**Mean Total Tracking Time [ms]**
00	27.36	141.14	28.34	145.4	61.07
01	28.27	105.33	31.53	105.21	70.85
02	27.13	117.74	28.61	117.01	60.53
04	28.38	138.56	30.07	138.93	61.24
05	28.52	129.78	29.99	137.95	62.04
06	29.29	150.26	31.18	150.41	63.81
07	27.82	91.89	28.4	95.34	58.35
09	27.74	95.75	28.5	98.17	58.7
10	26.69	78.5	27.85	81.8	57.23

## Data Availability

Publicly available datasets, KITTI, were analyzed in this study. This data can be found here: http://www.cvlibs.net/datasets/kitti (accessed on 26 December 2022).

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
