# Peer review of "An Adaptive ORB-SLAM3 System for Outdoor Dynamic Environments"

_sensors, 2023, doi:10.3390/s23031359_

Round 1
Reviewer 1 Report
This study aims to address the issue of low accuracy in dynamic outdoor environments, and propose an adaptive feature point selection system for outdoor dynamic environments. The questions studied are interesting, well-structured, and clearly motivated, but the current version of the manuscript has the following issues to be addressed.
1. Please describe the similarities and differences between this paper and reference [a] in detail, and make experimental comparison.
2. Absence of time expenditure comparison experiment.
3. Lack of relevant experiments on the quantity threshold of prior objects.
4. In the introduction part, it is suggested to make a separate paragraph and summarize the contributions of the article.
5. It is recommended to have a separate section detailing the relevant work and a table summarizing the strengths and weaknesses of these works, which will help to better understand the scope and novelty of the author's work.
6. In Section 2.1, it is suggested to use formal symbolic language to describe the system process.
7. The proposed method and experiment are described separately, and the experiment and analysis are conducted in a separate section.
8. Each symbol in formulas 1 and 2 needs a corresponding definition.
9. The table needs to be self-explanatory. The first column of the table is missing the header.
[a]. Zhang, Xinguang & Zhang, Ruidong & Wang, Xiankun. (2022). Visual SLAM Mapping Based on YOLOv5 in Dynamic Scenes. Applied Sciences. 12. 11548. 10.3390/app122211548.
Author Response
Dear Reviewers,
We feel great thanks for your professional review work on our article.
As you are concerned, there are several problems that need to be addressed. According to your nice suggestions, we have made extensive corrections to our previous draft. These changes will not influence the content and framework of the paper. The changes we made to the paper are visible in the pdf "paper with changes tracking", with additions marked in blue and deletions marked in red. The pdf "paper(Revisions)" is the revised paper. The detailed corrections are listed below:
- Comment: Please describe the similarities and differences between this paper and reference [a] in detail, and make experimental comparison.
Response: Thank you for your introduction to this wonderful research work. Reference [a] is a very innovative and well-structured article and we describe reference [a] in detail in our article (line 68, Table 1). Compared with reference [a], we both use YOLOv5 and LK optical flow methods for dynamic region determination, and both use the eight-point method. However, the specific method is different. Firstly, the reference [a] obtained the matching points by optical flow method tracking and then calculated the relationship from the feature points to the polar lines to determine the dynamic feature points, we pass the optical flow method to obtain the matching points and then used RANSAC to solve the essential matrix, and calculated the infinite norm of the essential matrix within the bounding box and bounding box expansion layer between two frames to determine the dynamic region. Furthermore, After obtaining the initial dynamic feature points, reference [a] selects the map points by local bundle set adjustment, while we eliminate potential a priori dynamic objects to reduce the localization error of the system by an adaptive feature point selector. We are sorry that it may be difficult for us to conduct a comparison experiment with reference [a] because our application scenarios are different. We enhance the detection of outdoors a priori by adding the coordinate attention mechanism to YOLOv5, our article is mainly applied to outdoor environments and we use the KITTI dataset for evaluation, reference [a] mainly uses the TUM indoor dynamic environment dataset for performance evaluation, and it is difficult for us to reproduce their system and conduct experiments in a short time.
- Comment: Absence of time expenditure comparison experiment.
Response: Thanks for your valuable counsel. We have compared the real-time performance of our system in detail (Table 9).
- Comment: Lack of relevant experiments on the quantity threshold of prior objects.
Response: We added experiments on thresholds for the number of a priori dynamic objects (Table 5).
- Comment: In the introduction part, it is suggested to make a separate paragraph and summarize the contributions of the article.
Response: We apologize for the poor language of our manuscript. We provide a detailed summary of the contributions of the article in the introduction section (line 95).
- Comment: It is recommended to have a separate section detailing the relevant work and a table summarizing the strengths and weaknesses of these works, which will help to better understand the scope and novelty of the author's work.
Response: We agree with this suggestion and have modified terminology throughout the text as appropriate. In the introduction section of this paper, a distinction has been made between relevant work and the contributions of our study. Additionally, a table has been incorporated to assist readers in understanding the relevant work(Table 1).
- Comment: In Section 2.1, it is suggested to use formal symbolic language to describe the system process.
Response: Thank you for pointing this out. We have used formal symbolic language to explain the system process in section 2.1.
- Comment: The proposed method and experiment are described separately, and the experiment and analysis are conducted in a separate section.
Response: We distinguish between the proposed approach and experiments, such as the performance experiments of YOLOv5 after introducing the coordinate attention mechanism. After distinguishing the experiment and the method we described the method in formal symbolic language to make the article more vivid.
- Comment: Each symbol in formulas 1 and 2 needs a corresponding definition.
Response: We apologize for the confusion generated by the previous version of the manuscript and sincerely hope that our logic is now easier to follow with this new version. We have added the definition of the symbols in the relevant formula.
- Comment: The table needs to be self-explanatory. The first column of the table is missing the header.
Response: We feel very sorry for our carelessness. We have added the title of the first column of the relevant table.
We would love to thank you for allowing us to resubmit a revised copy of the manuscript and we highly appreciate your time and consideration. Should you have any questions, please contact us without hesitation.
Kind regards,
Corresponding author Prof. Dr. Kehua Zhang
Email: zhangkh207@zjnu.edu.cn
Author Mr. Qiuyu Zang
Email: zangqiuyu@zjnu.edu.cn
References:
- Zhang, Xinguang & Zhang, Ruidong & Wang, Xiankun. (2022). Visual SLAM Mapping Based on YOLOv5 in Dynamic Scenes. Applied Sciences. 12. 11548. 10.3390/app122211548.

Reviewer 2 Report
The authors have studied the Simultaneous Localization and Mapping (SLAM) for the outdoor dynamic system performance. In this study, authors used Tracking, Local Mapping, and Local Closing threads. The system and methodology adopted was discussed well in the manuscript. They used YOLOv5s Detection thread and LK optical flow thread on top of ORB-SLAM3. This research presented the earlier study carried out in dynamics system and it may be in a separate section. The manuscript is prepared well and suitable for publication in the Sensors journal. Hence, it may be accepted with the following changes incorporated in the manuscript.
· Authors have claimed in the abstract that their system is more accurate in dynamic environment than others system (refer line number 11), without any quantitative results how the statement is correct, i.e. in what percentage your system performance is better. Similarly, in the conclusions the numerical output of the research finding have to be incorporated.
· Typographic errors must be solved (in abstract line number 9 no space after full stop, in line number 26 without space etc.). Throughout the manuscript need for correction in the space between words.
· The full form of KITTI - Karlsruhe Institute of Technology and Toyota Technological Institute required to be included at first instant or in the list of abbreviations. Check all the short forms and include in the abbreviation list
· In this research, authors have set the threshold to one pixel, why it is one pixel, explanation required.
· Table 5 is not cited and discussed in the manuscript, include it
Author Response
Dear Reviewers,
Thank you for your comments concerning our manuscript.
These comments are all valuable and very helpful for revising and improving our paper, as well as the important guiding significance to our research. We have studied the comments carefully and have made corrections which we hope meet with approval. The changes we made to the paper are visible in the pdf "paper with changes tracking", with additions marked in blue and deletions marked in red. The pdf "paper(Revisions)" is the revised paper. The responses to the comments are as follows:
- Comment: Authors have claimed in the abstract that their system is more accurate in dynamic environment than others system (refer line number 11), without any quantitative results how the statement is correct, i.e. in what percentage your system performance is better.Similarly, in the conclusions the numerical output of the research finding have to be incorporated.
Response: We greatly appreciate the valuable feedback provided. We have incorporated quantitative data into the manuscript, We have included the quantitative results 3 in the Abstract(line 11), Results section (line 291, line 301, etc.), and the conclusion section (line 332, line 360, etc.). Experiments show that our proposed system demonstrates a reduction in both absolute trajectory error and relative trajectory error, with a maximum reduction of 39% and 30% respectively compared to other systems.
- Comment: Typographic errors must be solved (in abstract line number 9 no space after full stop, in line number 26 without space etc.). Throughout the manuscript need for correction in the space between words.
Response: Thank you for pointing out this problem. We have made a diligent attempt to address and resolve any typographic errors present within the article.
- Comment: The full form of KITTI - Karlsruhe Institute of Technology and Toyota Technological Institute required to be included at first instant or in the list of abbreviations. Check all the short forms and include in the abbreviation list.
Response: We apologize for the language problems in the original manuscript. An examination of all shortened forms was conducted and subsequently incorporated into the abbreviation list, For example, KITTI, VINS, TSDF, etc.
- Comment: In this research, authors haveset the threshold to one pixel, why it is one pixel, explanation required.
Response: We are very sorry for our negligence in the absence of the explanation of the threshold. An experiment on the impact of the distance from the matching point to the epipolar lines in the RANSAC algorithm resulted in this threshold. Specifically, we respectively evaluated various thresholds on the dataset containing a large number of dynamic objects and a large number of static objects and discovered that the system achieved a reduction in the absolute trajectory error at a threshold of 1 pixel. Our findings were subsequently included in the manuscript (Table 4).
- Comment: Table 5 is not cited and discussed in the manuscript, include it.
Response: We apologize for the lack of citations in the form and thank you for pointing this out.
Kind regards,
Corresponding author Prof. Dr. Kehua Zhang
mail: zhangkh207@zjnu.edu.cn
Author Mr. Qiuyu Zang
Email: zangqiuyu@zjnu.edu.cn

Round 2
Reviewer 1 Report
This version addresses my concerns about the previous version, and my recommendation is to accept it. Congratulations.
Author Response
Dear Reviewer,
We feel great thanks for your professional review work on our article. Thank you very much for your attention and consideration. Thank you for your contribution to our manuscript.
Kind regards,
Corresponding author Prof. Dr. Kehua Zhang
Email: zhangkh207@zjnu.edu.cn
Author Mr. Qiuyu Zang
Email: zangqiuyu@zjnu.edu.cn
